# Mesoscale eddies heterogeneously modulate CO<sub>2</sub> fluxes in eddy-rich regions of the Southern Ocean

Mariana Salinas-Matus<sup>1,2,3</sup>, Nuno Serra<sup>3</sup>, Fatemeh Chegini<sup>3</sup>, and Tatiana Ilyina<sup>3,4,1</sup>

**Correspondence:** Mariana Salinas-Matus (mariana.salinas@mpimet.mpg.de)

Abstract. Mesoscale eddies are known to influence the Southern Ocean biogeochemistry. However, the distinct contributions of cyclonic and anticyclonic eddies to air-sea  $CO_2$  fluxes, as well as their longer-term effects remain poorly studied. We present results from a 27-year global eddy-resolving ocean-biogeochemical simulation. We used the Okubo-Weiss parameter to classify the modeled flow regimes into cyclonic and anticyclonic eddies, peripheries, and the surrounding background waters. Our results reveal a heterogeneous influence of eddies depending on the region, driven by regional differences in eddy intensity and the gradients in background properties. The factors controlling  $CO_2$  fluxes within eddies follow the same degree of importance as in background waters, with  $\Delta pCO_2$  being the dominant factor. This is driven primarily by changes in dissolved inorganic carbon. Our analysis shows that eddies act as a persistent carbon sink on decadal timescales, while their influence on shorter timescales is more variable and strongly shaped by eddy polarity. Overall, mesoscale regimes exhibit higher carbon uptake efficiency, with anticyclonic eddies showing the highest efficiency. The ability of eddies to absorb carbon computed in our results is consistent with recent observational estimates, confirming that the model realistically represents the influence of mesoscale eddies on  $CO_2$  fluxes. Above all, the overall contribution of mesoscale eddies to carbon uptake across the Southern Ocean was relatively small, accounting for approximately 10% of total and 1% of anomalous carbon uptake. However, the regional influence is more pronounced in eddy-rich regions.

#### 15 1 Introduction

The Southern Ocean exhibits high mesoscale eddy activity (Chelton et al., 2011b; Petersen et al., 2013). These mesoscale eddies influence the physical properties of seawater by transporting heat and salinity across regions and into the ocean's interior (Morrison et al., 2013; Griffies et al., 2015; Morrison et al., 2016; von Storch et al., 2016). Similarly, these eddies play a substantial role in the transport of biogeochemical properties (Chelton et al., 2011a; Dufour et al., 2015; Schütte et al., 2016b; Moreau et al., 2017; Dawson et al., 2018; Patel et al., 2020; Su et al., 2021; Lévy et al., 2023). Notably, the mass transport induced by eddies is comparable to large-scale transport (Zhang et al., 2014). Moreover, eddies play a significant role in shaping Southern Ocean future climate projections (Farneti et al., 2010; Dufour et al., 2015; Hogg et al., 2015; Bilgen

<sup>&</sup>lt;sup>1</sup>Ocean Biogeochemistry, Climate Variability Department, Max Planck Institute for Meteorology, Hamburg, Germany

<sup>&</sup>lt;sup>2</sup>International Max Planck Research School on Earth System Modelling, Hamburg, Germany

<sup>&</sup>lt;sup>3</sup>Modeling the carbon cycle in the Earth system, Department of Earth System Sciences, Universität Hamburg, Hamburg, Germany

<sup>&</sup>lt;sup>4</sup>Earth system modeling, Helmholtz-Zentrum Hereon, Geesthacht, Germany

and Kirtman, 2020; van Westen and Dijkstra, 2021; Putrasahan et al., 2021). However, the study of mesoscale eddies has been limited by the lack of biogeochemical datasets that can capture the high eddy activity in the region, both from observational and modeling perspectives. Observational studies are limited by the sparse spatial and temporal coverage of measurements (Gray, 2023; Dong et al., 2024), while eddy-resolving simulations are limited by their high computational cost (Hewitt et al., 2022; Guo and Timmermans, 2024).

Despite growing research interest, the role of cyclonic and anticyclonic eddies in modulating air-sea  $CO_2$  fluxes remains inconclusive. Regional and short-term oceanographic campaigns have identified that mesoscale eddies exhibit diverse influence on carbon uptake. It has been shown that both cyclonic and anticyclonic eddies can function as carbon sinks (Jones et al., 2017; Orselli et al., 2019a; Ford et al., 2023). Conversely, other studies have reported inverse patterns, where anticyclonic eddies release  $CO_2$  to the atmosphere, while cyclonic eddies serve as carbon sinks (Pezzi et al., 2021; Kim et al., 2022). Additionally, it has been shown that there are seasonal differences in the carbon uptake depending on the eddy polarity (Song et al., 2016; Jones et al., 2017).

35

50

Only recently have studies begun to investigate the effects of mesoscale processes on air-sea CO<sub>2</sub> fluxes at larger spatial scales (Keppler et al., 2024; Guo and Timmermans, 2024; Li et al., 2025). Keppler et al. (2024) combined mesoscale eddies identified from satellite altimeter with data from biogeochemical Argo floats, showing on average that both cyclonic and anticyclonic eddies contribute to the carbon uptake in the Southern Ocean. However, cyclonic eddies were associated with a weaker carbon uptake or more outgassing, whereas anticyclonic eddies were associated with stronger carbon uptake or less outgassing. Li et al. (2025), using an observation-based machine learning approach, showed that anticyclonic eddies substantially enhance CO<sub>2</sub> uptake on average, whereas cyclonic eddies contribute less consistently. From a modeling perspective, Guo and Timmermans (2024) used an eddy-resolving global biogeochemical simulation to isolate the effect of mesoscale activity. Their findings indicate that mesoscale dynamics account for over 30 % of the total variance of the air-sea CO<sub>2</sub> fluxes in energetic regions and can act either as sources or sinks depending on the region, showing the importance of mesoscale dynamics from a global perspective. From a regional modelling perspective in the South–East Atlantic Ocean, studies have shown distinct thermodynamic and biogeochemical signatures between cyclonic and anticyclonic eddies (Smith et al., 2023), as well as a high occurrence of eddies with abnormal characteristics that can weaken or even reverse the air-sea fluxes (Smith et al., 2025). However, no modeling study to date has investigated the contrasting impacts of cyclonic and anticyclonic eddies on air-sea CO<sub>2</sub> fluxes across the entire Southern Ocean, while also considering regional differences.

This distinction may be important because cyclonic and anticyclonic eddies possess contrasting physical structures that are expected to influence air-sea CO<sub>2</sub> fluxes differently. Anticyclonic eddies are high-pressure systems characterized by positive vorticity in the southern hemisphere, which causes isopycnals to be pushed downward. This induces water downwelling, resulting in higher surface temperatures compared to surrounding waters, a mechanism known as eddy-pumping. Higher temperature reduces the solubility of CO<sub>2</sub> but the downwelling effect may transport carbon to deeper ocean layers. The opposite pattern occurs in cyclonic eddies, which are characterized by negative vorticity in the southern hemisphere, uplifted isopycnals, and water upwelling (McGillicuddy, 2016; Jones et al., 2017; Ford et al., 2023).

The regional differences within the Southern Ocean add another layer of complexity when attempting to generalize eddy behavior (McGillicuddy, 2016). For instance, the Agulhas Current system, south of Africa, is highly energetic and contributes significantly to the global thermohaline circulation via eddies and filaments involved in the Agulhas leakage (Richardson, 2007; Biastoch et al., 2009; Beal et al., 2011; Haarsma et al., 2011). The Brazil-Malvinas Confluence Zone, located where the Brazil and Malvinas Currents meet, is characterized by strong temperature and salinity gradients, which create high-energy eddies and thermohaline fronts (Barré et al., 2006; Mason et al., 2017; Souza et al., 2021). In contrast, the region South of Tasmania is mostly influenced by the Antarctic Circumpolar Current (ACC), which produces a complex frontal system and robust eddy activity, influencing the water mass exchange between major ocean basins and the deep ocean (Rintoul and Sokolov, 2001). These distinct regional settings suggest that eddies impacts on CO<sub>2</sub> fluxes may not be uniform across the Southern Ocean, highlighting the need to investigate how mesoscale eddies operate in different oceangraphic contexts.

Our objective in this work is to determine how anticyclonic and cyclonic eddies, periphery, and the surrounding background waters modulate air-sea  $CO_2$  fluxes, to identify the physical and biogeochemical mechanisms driving these fluxes, and to examine how these mechanisms vary across different eddy-rich regions and the entire Southern Ocean. To achieve this, we use a global eddy-resolving ocean-biogeochemical model. Our model simulation is long enough to study the influence of eddies on  $CO_2$  fluxes across timescales ranging from daily to decadal.

#### 2 Methods

70

## 2.1 Model description

We use the ocean model ICON-O (Korn, 2017; Korn et al., 2022) incorporating the ocean biogeochemistry model HAMOCC (Six and Maier-Reimer, 1996; Ilyina et al., 2013). ICON-O solves the hydrostatic Boussinesq equations of the large-scale ocean dynamics with a free surface. The horizontal grid is composed of triangular cells with an Arakawa C-type staggering of variables. The grid is generated by subdividing the 20 triangular faces of an icosahedron inscribed into the sphere. The model computes tracer advection using a flux-corrected algorithm, which combines a low-order upwind scheme for stability with a higher-order scheme for accuracy. Vertical mixing is parametrized using a prognostic equation for turbulent kinetic energy, where the mixing efficiency is determined by a variable mixing length that adapts to local ocean conditions (Korn, 2017; Korn et al., 2022).

HAMOCC computes biological and chemical sources and sinks of biogeochemical tracers in the water column and sediment (Six and Maier-Reimer, 1996; Ilyina et al., 2013). Primary producers are represented by bulk phytoplankton and cyanobacteria, with growth limited by temperature, light, and nutrient availability. Grazing by zooplankton is restricted to bulk phytoplankton. Organic matter is divided into sinking particulate detritus and dissolved organic matter. All marine organic compounds are assumed to share a uniform elemental composition, based on a modified Redfield ratio. The biogeochemical tracers are advected and diffused by the ocean model. The ocean model also provides temperature, salinity and sea ice concentration to compute transformation rates, constants and fluxes. Atmospheric pressure, wind speed, and shortwave radiation are prescribed.

The air-sea exchange of CO<sub>2</sub> depends on the surface layer piston velocity (Wanninkhof, 2014), solubility factor (Weiss, 1974), and pCO<sub>2</sub> (See Appendix A). The pCO<sub>2</sub> is determined from the total dissolved inorganic carbon (DIC), total alkalinity, temperature, and salinity. Total DIC is taken as the sum of all dissociated inorganic carbon species concentrations. Changes of DIC are driven by photosynthesis, grazing, zooplankton excretion, remineralization of dissolved organic matter and detritus, carbonate calcium production and dissolution, and air-sea flux.

#### 2.2 Experiment design

We performed 27 years (1995-2022) of global simulations, saving daily average output. The nominal grid resolution varies between 8.4 and 10 km. The vertical grid is structured and non-uniform. We use 72 vertical layers, the thickness of the layers growing gradually with depth. The upper 300 m are represented by 29 layers. ERA5 is used as atmospheric forcing (Hersbach et al., 2020). The atmospheric pCO<sub>2</sub> used in the model was prescribed from the global CO<sub>2</sub> concentration dataset prepared by the Global Carbon Project (Friedlingstein et al., 2022). As this is an ocean-only configuration without coupling to the atmosphere, eddy-induced feedbacks on surface winds are not represented.

Due to the substantial computational time required for spin-up at high resolution and the complexity of HAMOCC, a direct spin-up at 10 km resolution was impractical. Therefore, a cascade strategy was implemented. First, the spin-up was conducted at a lower resolution of 40 km. A long spin-up of approximately 3,000 years was performed for the ocean's physical conditions with ICON-O only. The spin-up for the HAMOCC biogeochemistry model at pre-industrial conditions was carried out for around 900 years, until the upper ocean drift stabilized. The 40km resolution model was then run for the transient period until the year 1990. The physical and biogeochemical results from the coarser resolution were then used to initialize the 10 km resolution model. To allow the ocean conditions to adapt, only the ocean model was run for 5 years. After this adaptation period, the biogeochemical conditions were added. A tuning of the biogeochemical components in the 10 km resolution model was performed before the production phase of the simulation. This cascade strategy ensures an adequate representation of upper ocean mesoscale and regional processes, which are the main focus of this study. Furthermore, all model drifts in the 10 km resolution control run remain sufficiently small in accordance with the CMIP6 protocol (Jones et al., 2016).

#### 2.3 Eddy detection method



To detect mesoscale eddies and deformation-dominated areas (Fig. 1b and 2a), we implemented an algorithm based on the Okubo-Weiss parameter (OW) (Okubo, 1970; Weiss, 1991). OW measures the relative dominance of rotation (vorticity) against deformation (strain):

$$OW = \left(\frac{\partial u}{\partial x} - \frac{\partial v}{\partial y}\right)^2 + \left(\frac{\partial v}{\partial x} + \frac{\partial u}{\partial y}\right)^2 - \left(\frac{\partial v}{\partial x} - \frac{\partial u}{\partial y}\right)^2$$
(1)
Vorticity

Four flow regimes were defined using OW: anticyclonic eddy cores, cyclonic eddy cores, periphery, and quiescent "background". The "background" regime is defined using a threshold equal to 0.3 of the temporal mean of the spatial standard

deviation of the OW ( $\sigma_{OW}$ ) corresponding to  $\pm 0.5 \times 10^{-10} s^{-2}$ , encompassing all OW values within this range. The most common approach in the literature is to define the threshold as  $0.2\sigma_{OW}$  (Schütte et al., 2016a; Vu et al., 2018), with some studies adopting more relaxed values around  $0.1\sigma_{OW}$  (Beech et al., 2025). However, in this study we chose to apply a slightly stricter criterion in order to isolate more robust and well-defined eddy structures (see Appendix C). Vorticity-dominated regions (eddy flow regimes) are identified as closed contours with negative OW values smaller than  $-0.5 \times 10^{-10} s^{-2}$ . Relative vorticity is then used to distinguish between anticyclonic and cyclonic eddy cores. Deformation-dominated regions, referred to here as the periphery regime, are defined as areas with positive OW values exceeding  $0.5 \times 10^{-10} s^{-2}$ . These regions occur at the interacting regions between eddies and between eddies and large-scale currents, or along meanders and filaments, where the flow is stretched or compressed rather than rotating. Figure 2a provides a snapshot illustrating the defined flow regimes.

Eddies were not tracked, so the daily eddy-covered areas may represent either the persistence of the same eddies or the appearance of new ones from one day to the next. Additionally, the eddy lifetime was not considered in the analysis. Composites of each flow regime were generated for studying the overall effect of the flow regimes on the CO<sub>2</sub> flux. Here, a composite refers to the area-weighted average of relevant variables across all identified instances of a given flow regime, providing a representative picture of the typical conditions associated with each flow regime.

The net CO<sub>2</sub> flux is influenced by processes operating across different timescales (Gu et al., 2023). To determine the frequency range at which mesoscale eddies have the greatest impact on CO<sub>2</sub> flux, we first applied a Fourier analysis to the full spatial variable fields to decompose it into distinct frequency bands. This approach ensures that the spatial and temporal coherence of the field is preserved before the identification of eddies. The analysis yielded five filtered fields corresponding to (1) intra-annual variations above 16 months and up to 27 years (M16), (2) annual variations between 8 and 16 months (M8–M16), (3) intra-annual variations between 4 and 8 months (M4–M8), (4) intra-annual variations between 1 and 4 months (M1–M4), and (5) submonthly variations shorter than 1 month, down to daily changes (M1). Subsequently, eddy composites were computed for each filtered field using the criterion described above.

#### 3 Results and discussion




## 3.1 Simulated mesoscale eddy characteristics

Our 10 km resolution simulation captures mesoscale eddy activity (Fig. 1a and b) in the Southern Ocean, which is defined here as the ice-free ocean south of 30°S. The model simulates high vorticity values (positive and negative) in the ACC region, the Agulhas Retroflection region, the Brazil-Malvinas Confluence region, and the coast and south of Australia. Those regions also present the largest vorticity variability, highlighting intense mesoscale dynamics (Fig. 1a). The detected eddies cannot be directly compared with eddies tracked from observations. However, our model's eddy counts are consistent with the distribution of eddy kinetic energy attributed to eddies reported by Chelton et al. (2011b). Moreover, the distribution of detected eddies aligns with the Southern Ocean's eddy hotspots reported by Frenger et al. (2015) (Fig. 1b).

The average daily number of eddies varies across regions, ranging from 11 to 38 eddies present each day. There is little to no difference between the number of anticyclonic and cyclonic eddies within each region. In terms of spatial coverage,

mesoscale flow regimes occupy between 10 - 20% of the regions (Table 1). Satellite observations suggest this field typically covers 25-30% of the ocean surface, regardless of the size of the region (Chaigneau et al., 2009). It is important to consider that the model has a limited ability to simulate the full eddy activity in the highest latitudes, where the deformation radii are well below 10 km.

**Table 1.** Summary of eddy statistics for each region. Mesoscale flow regimes include anticyclonic eddies, cyclonic eddies, and periphery. For the entire Southern Ocean, mesoscale flow regimes cover 9.31% of the total area of  $106.1 \times 10^6$  km<sup>2</sup>.

| Region                     | $\begin{array}{ccc} & & & \text{Area covered} \\ \text{total area} & \text{by mesoscale flow regimes} \\ (10^6 \text{ km}^2) & \% & \end{array}$ |      | Daily number of eddies |          |
|----------------------------|--------------------------------------------------------------------------------------------------------------------------------------------------|------|------------------------|----------|
|                            |                                                                                                                                                  |      | Anticyclones           | Cyclones |
| Agulhas Retroflection      | 14.3                                                                                                                                             | 16.7 | 38                     | 32       |
| Brazil-Malvinas Confluence | 10.2                                                                                                                                             | 9.9  | 14                     | 11       |
| South Tasmania             | 4.8                                                                                                                                              | 19.7 | 23                     | 22       |



Based on eddy spatial density in the model, we define three eddy-rich regions: the Brazil-Malvinas Confluence region, the Agulhas Retroflection region and the South of Tasmania region (Fig. 1b, orange rectangles). The selection is made due to the heterogeneous nature of the Southern Ocean which makes it difficult to generalize the effect of eddies in the CO<sub>2</sub> flux. For instance, the eddy intensity differs between regions and is expected to modulate air—sea CO<sub>2</sub> fluxes, as stronger eddies enhance both lateral and vertical transport of water properties, including temperature and DIC. The Agulhas Retroflection region presents the strongest and most persistent anticyclonic and cyclonic eddy intensity. Eddies in the Brazil-Malvinas Confluence region are less intense but with higher variability. Anticyclones in the Tasmania region are the least intense among the three regions but with the smallest variability, while cyclonic intensity is similar to the Brazil-Malvinas Confluence region but with persistent intensity (Fig. 1c). Frenger et al. (2015) showed that the intensity is stronger in the Agulhas Retroflection, followed by the Brazil-Malvinas Confluence, and weakest south of Tasmania.

#### 3.2 Heterogeneous dissolved inorganic carbon and temperature patterns across flow regimes in eddy-rich regions

The selected regions exhibit distinct characteristics, influenced by their latitudinal positions, ocean current, and frontal systems, and mesoscale activity. In the Agulhas Retroflection and Brazil-Malvinas Confluence regions, both situated at similar latitudes, a pronounced spatial pattern emerges (Fig. 3a). The pronounced spatial pattern is marked by warmer temperatures, higher salinity, and lower pCO<sub>2</sub> and DIC concentrations to the north, contrasting sharply with colder, fresher, and carbon-rich waters to the south. The Tasmania region, located farther south, experiences water upwelling that brings carbon-rich waters from the deep to the surface, resulting in colder temperatures and high DIC concentrations throughout the year (Fig. 3a). The overall behavior of the simulated temperature and DIC is consistent with the oceanographic knowledge of the Southern Ocean (Talley et al., 2011). Mesoscale activity plays a substantial role in modulating carbon and temperature patterns (Fig. 2b-d).

Figure 1. Overview of modeled mesoscale eddies in the simulation period (1996–2022): (a) Mean and standard deviation of vorticity, with an approximate location of the ACC, plotted based on surface height, (b) Locations of detected anticyclonic and cyclonic eddy centroids, defined regions (orange rectangles), and eddy centroids census within  $1^{\circ} \times 1^{\circ}$  grid cells, and (c) Relative vorticity composites for anticyclonic and cyclonic eddies within selected regions.

In our simulations, flow regimes exhibit distinct characteristics that influence the CO<sub>2</sub> flux (Fig. 2b-d), with contrasting vertical structures of cyclonic and anticyclonic eddies emerging (Fig. 3b), consistent with previous findings (Keppler et al., 2024). Anticyclones, which tend to induce water downwelling, exhibit higher temperatures and lower DIC concentrations. In contrast, cyclones exhibit lower temperatures and higher DIC concentrations, mostly driven by water upwelling. The periphery regime shows intermediate properties, representing the transition zones surrounding both anticyclonic and cyclonic eddies (Fig. 3b). While this general pattern is evident across all three regions, the magnitude of the differences varies with latitude and local dynamics, reflecting the influence of regional circulation.

The strong relationship between temperature, DIC, and gas solubility directly influences the exchange of  $CO_2$  between the ocean and the atmosphere. The Southern Ocean's overall  $CO_2$  uptake capacity is evident in our simulation. An outgassing band between  $40^{\circ}S$  and  $55^{\circ}S$ , particularly in the Atlantic and Indian sectors, is associated with the high DIC concentration in the Southern Ocean upwelling region. North of  $60^{\circ}S$ , the  $CO_2$  flux exhibits relatively low variability compared to the polar region but with notable localized signals influenced by mesoscale structures (Fig. 3c).

**Figure 2.** Snapshot of the identified mesoscale dynamics and tracer or tracer flux anomalies in the Agulhas Retroflection region (defined in Fig. 1b) on 03 June 2014. (a) Okubo–Weiss parameter, delineating three flow regimes: vorticity-dominated (green), deformation-dominated (violet), and "background" (grey). (b) Sea Surface Temperature (SST) anomaly, (c) air–sea CO<sub>2</sub> flux anomaly, and (d) Surface DIC anomaly. Anomalies represent deviations from the climatological state, computed as the difference between the instantaneous value and the long-term mean at each grid point. The air–sea CO<sub>2</sub> flux is defined as positive for outgassing (flux from the ocean to the atmosphere) and negative for uptake (flux from the atmosphere to the ocean). Therefore, a positive flux anomaly indicates increased outgassing relative to the mean, while a negative anomaly indicates increased uptake relative to the mean. In all panels, contours outline anticyclonic and cyclonic eddies.

In the Agulhas Retroflection and Brazil-Malvinas Confluence regions, all four regimes show a mean distribution indicating carbon uptake, with a wider spread toward stronger uptake in both cyclonic and anticyclonic eddies. However, the distribution of cyclonic eddies and "background" shifts slightly toward carbon outgassing events (Fig. 3c).

In contrast, the Tasmania region presents a different pattern. The "background" shows a mean data distribution close to zero CO<sub>2</sub> flux, while cyclonic, anticyclonic, and periphery regimes indicate carbon release. Cyclonic eddies exhibit the highest outgassing. However, all regimes present a broad spread extending towards higher carbon uptake (Fig. 3c). The outgassing pattern might be associated with the predominance of eddies in zones influenced by upwelling carbon-rich waters (Fig. 3a) and the overall lower eddy intensity in the region (Fig. 1c).

#### 3.3 Enhanced CO<sub>2</sub> uptake by anticyclonic eddies




The integrated CO<sub>2</sub> flux over the 27 analyzed years indicates that, in most regions, mesoscale flow regimes (anticyclones, cyclones, and periphery) exhibit a greater efficiency in carbon uptake compared to the "background", although this pattern is not consistent across the entire Southern Ocean (Fig. 4a). When accounting for the area they occupy, these regimes are responsible for 15–22% of the carbon uptake within the defined regions, and 10% of the total carbon uptake in the entire Southern Ocean (Table 2). However, when considering the anomalous contribution, i.e., the additional or reduced carbon uptake relative to "background" conditions, the percentage contribution decreases substantially, depending on the local efficiency of the mesoscale structures. This anomalous contribution is largest in the Agulhas Retroflection region, where mesoscale regimes enhance uptake by about 10%, while in the South Tasmania region, the flux is approximately 17% lower than under

**Figure 3.** Overview of sea surface temperature (SST), surface DIC, and CO<sub>2</sub> flux in the Southern Ocean and composite quantities for flow regimes within the defined regions: (a) Mean and standard deviation of SST and DIC; (b) Vertical temperature and DIC profile composites for the flow regimes within the defined regions, (c) Mean and standard deviation of CO<sub>2</sub> flux, along with composite CO<sub>2</sub> fluxes for the flow regimes within the defined regions.

"background" conditions. Across the entire Southern Ocean, the anomalous contribution is strongly reduced to less than 1%. This low relative value primarily reflects the large spatial extent of the domain, where the "background" conditions dominate and contribute substantially to the total carbon uptake, thereby diluting the relative impact of mesoscale regimes (Table 2).



It is noteworthy that overall anticyclonic eddies show higher efficiency in taking up CO<sub>2</sub> (Fig. 4a). In particular, in the Agulhas Retroflection and Brazil-Malvinas Confluence regions, anticyclonic eddies exhibit the highest carbon uptake capacity, whereas the "background" has the lowest carbon uptake per unit area. In these two regions, cyclonic eddies also exhibit an enhanced ability to take up carbon compared with the "background", although not as intense as anticyclonic eddies (Fig. 4a). Anticyclonic and cyclonic eddies have the ability to take up more, or outgas less, carbon compared to the "background" (Fig. 4c). This is explained by the stronger eddy-intensity (Fig. 1c) and the stronger vertical gradient in those two regions, as indicated by the temperature and DIC profiles (Fig. 3b).

Tasmania region exhibits a contrasting pattern, as the "background" has the highest CO<sub>2</sub> uptake (Fig. 4a). Here, both anticyclonic and cyclonic eddies take up less or outgas more carbon (Fig. 4c). This pattern is influenced by the weaker eddy intensity (Fig. 1c) and the characteristics of the region. Most eddies in the Tasmania region are located in the high-DIC band, which limits the ocean's capacity for CO<sub>2</sub> uptake (Fig. 3b). Nevertheless, anticyclonic and cyclonic eddies take up carbon in the entire analyzed period (Fig. 4a), a pattern influenced by sporadic but intense carbon uptake events occurring across all flow regimes in the region (Fig. 3c). These events may be associated with periods of reduced upwelling activity, which limit the vertical transport of DIC-rich subsurface waters to the surface, thereby allowing enhanced CO<sub>2</sub> uptake (Pardo et al., 2017). However, since the efficiency of CO<sub>2</sub> uptake in mesoscale flow regimes is lower than in the "background", the anomalous contribution is negative. Mesoscale regimes take up approximately 17% less carbon compared to what would be expected under "background" conditions (Table 2).







The periphery regime displays an intermediate carbon uptake between anticyclonic and cyclonic eddies. However, when considering the area it covers, the periphery regime is the mesoscale regime that contributes the most to net CO<sub>2</sub> uptake, accounting for around 60–70% of the carbon uptake generated by the mesoscale flow regimes, and a comparable percentage of the anomalous contribution (Table 2). Given this substantial contribution, understanding the physical and biogeochemical drivers of CO<sub>2</sub> uptake in the periphery regime is an important direction for future research. Meanwhile, the combined contribution of anticyclonic and cyclonic eddies represents about 20–30% of the mesoscale regimes' carbon uptake, and a similar proportion of the anomalous contribution (Table 2). This is particularly relevant given the observed trend of increasing eddy kinetic energy (EKE) in the Southern Ocean over recent decades (Hogg et al., 2015). If this trend continues, it could lead to a more eddy-active Southern Ocean, with an expansion of the area influenced by mesoscale features. Such changes may potentially enhance or reduce the ocean's capacity for CO<sub>2</sub> uptake in the future depending on the region, as larger regions become dominated by eddies.

The influence of mesoscale eddies on CO<sub>2</sub> flux has been identified through in situ data. Previous studies have shown that Agulhas anticyclonic eddies have a stronger CO<sub>2</sub> uptake ability than their surrounding water (Orselli et al., 2019b), which contributes to the faster acidification of the region (Orselli et al., 2019a). In the Tasmania region, Jones et al. (2017) identified eddies as hotspots for carbon uptake. In the Brazil-Malvinas Confluence region, a single anticyclonic eddy was identified as a CO<sub>2</sub> source to the atmosphere (Pezzi et al., 2021).

Our model results are generally consistent with these observational findings. We observe a notably enhanced carbon uptake by anticyclones in the Agulhas region compared to the other regions, and eddies in the Tasmania region also act as persistent CO<sub>2</sub> sinks. In the Brazil–Malvinas Confluence, a direct comparison is not possible since our analysis focuses on the overall eddy effect over time rather than on individual eddy events. However, it is worth mentioning that both anticyclonic and cyclonic eddies do exhibit episodes of CO<sub>2</sub> outgassing, although such events are more frequently in cyclonic eddies (Fig. 3c).

Despite regional heterogeneity, anticyclonic eddies take up more or outgas less carbon than the "background", while cyclonic eddies exhibit the opposite pattern (Fig. 4c). This is consistent with studies using observation-based datasets (Keppler et al., 2024; Li et al., 2025). This contrasting behavior may be attributed to eddy-pumping. In anticyclonic eddies, this mechanism facilitates the transfer of carbon to deeper ocean layers, enhancing the ocean's capacity for carbon uptake over time. In contrast,

cyclonic eddies may act to reduce carbon uptake by limiting this downward transport. While this opposition leads to a partial compensation between positive and negative flux anomalies over the entire Southern Ocean, such compensation is not observed at the regional scale, where both anticyclonic and cyclonic eddies exhibit flux anomalies of the same sign (Table 2).



The strong agreement between the model and observations underscores the critical role of mesoscale eddies in modulating CO<sub>2</sub> fluxes. This agreement indicates that the model is effectively capturing the key physical and biogeochemical processes involved. Moreover, the model provides a more detailed and continuous representation of eddy dynamics, offering a more robust signal than observations alone, which may be spatially or temporally limited.

**Figure 4.** (a) Time-integrated CO<sub>2</sub> flux composites for flow regimes across the defined regions, including relative contributions of CO<sub>2</sub> flux drivers (solubility, pCO<sub>2</sub> and, piston velocity) to the variability. (b) For the Agulhas Retroflection region, though the other regions follow the same pattern, relative contributions of these drivers across different frequency bands (M16: interannual variations above 16 months, M8-M16: annual variations between 8 and 16 months, M4-M8: intra-annual variations between 4 and 8 months, M1-M4: intra-annual variations between 1 and 4 months, and M1: submonthly variations). (c) Composite anomalies of total CO<sub>2</sub> flux relative to the "background" for anticyclonic and cyclonic eddies. (d–f) Composite anomalies of CO<sub>2</sub> flux across different frequency bands in three regions: (d) Agulhas Retroflection, (e) Brazil–Malvinas Confluence, and (f) south of Tasmania.

**Table 2.** Area-integrated total and anomalous carbon uptake contributions by region and mesoscale flow regime. The "Total" rows correspond to the absolute uptake, and the "Anomalous" rows show the deviation relative to "background" conditions. Values in parentheses denote the contribution as a percentage, first relative to the total uptake (in bold), and second relative to the combined uptake from all mesoscale flow regimes (eddies + periphery). All values are mean  $\pm$  standard deviation.

## Carbon uptake [Pg C yr<sup>-1</sup>]

| Region                                       | Туре      | Mesoscale total                         | Disentangling Mesoscale flow regimes       |                                                                           |                                         |
|----------------------------------------------|-----------|-----------------------------------------|--------------------------------------------|---------------------------------------------------------------------------|-----------------------------------------|
|                                              | lype      |                                         | Anticyclones                               | Cyclones                                                                  | Periphery                               |
| Agulhas<br>Retroflection<br><b>0.14±0.08</b> | Total     | $0.032 \pm 0.016$ (22 $\pm 11\%$ )      | $0.009 \pm 0.003$<br>(27 ± 11%)            | $0.003 \pm 0.002 \\ (10 \pm 7\%)$                                         | $0.020 \pm 0.010$<br>(62 ± 31%)         |
|                                              | Anomalous | $0.0115 \pm 0.009$<br>(10 $\pm 8\%$ )   | $0.00428 \pm 0.00030$ $(37 \pm 2\%)$       | $0.00056 \pm 0.00134$ $(5 \pm 12\%)$                                      | $0.00614 \pm 0.00507$ $(53 \pm 44\%)$   |
| Brazil-Malvinas<br>Confluence<br>0.10±0.07   | Total     | $0.013 \pm 0.009$ (13 $\pm$ 9%)         | $0.002 \pm 0.001$ $(15 \pm 7\%)$           | $0.001 \pm 0.001$ $(7 \pm 7\%)$                                           | $0.010 \pm 0.007 \\ (74 \pm 51\%)$      |
|                                              | Anomalous | $0.00348 \pm 0.006$<br>(4±0.06%)        | $0.00091 \pm 0.00138$<br>(26 ± 39%)        | $0.00017 \pm 0.00080$ $(5 \pm 23\%)$                                      | $0.00240 \pm 0.00388$<br>(69 ± 111%)    |
| South<br>Tasmania<br>0.02±0.05               | Total     | $0.003 \pm 0.010$ (15±50%)              | $0.0006 \pm 0.001$<br>(20 ± 62%)           | $0.0002 \pm 0.001$<br>$(8 \pm 60\%)$                                      | $0.002 \pm 0.006$<br>$(71 \pm 206\%)$   |
|                                              | Anomalous | $-0.00298 \pm 0.002$<br>(-17 $\pm$ 11%) | $-0.00059 \pm 0.00053$<br>$(-20 \pm 18\%)$ | $-0.00074 \pm 0.00061$ $(-25 \pm 20\%)$                                   | $-0.00165 \pm 0.00110$ $(-55 \pm 36\%)$ |
| Entire<br>Southern Ocean<br>1.17±0.60        | Total     | $0.118 \pm 0.063$ (10±5%)               | $0.020 \pm 0.010$<br>(18 ± 9%)             | $0.010 \pm 0.008$<br>(9 ± 7%)                                             | $0.086 \pm 0.045 \\ (72 \pm 38\%)$      |
|                                              | Anomalous | $0.00946 \pm 0.027$<br>(1 $\pm$ 2.5%)   | $0.00313 \pm 0.00531$<br>$(33 \pm 56\%)$   | $ \begin{array}{c c} -0.00136 \pm 0.00414 \\ (-14 \pm 43\%) \end{array} $ | $0.00769 \pm 0.0180$<br>(81 ± 190%)     |

The frequency analysis suggest that eddies act as a persistent carbon sink on decadal timescales (M16) (Fig. 4d-f). In the Agulhas Retroflection and Brazil-Malvinas Confluence regions, both anticyclonic and cyclonic eddies exhibit enhanced CO<sub>2</sub> uptake relative to the "background" (Fig. 4d and e). In contrast, eddies in the Tasmania region do not absorb more carbon than the "background" (Fig. 4f). Nevertheless, since the integrated CO<sub>2</sub> flux in this region remains an overall carbon uptake, these eddies persistently take up carbon despite the high concentrations of DIC in that region. This is particularly important since the recent trend has proposed increased eddy activity in the Southern Ocean (Hogg et al., 2015; Martínez-Moreno et al., 2021) and the growing influence of eddies on heat distribution in a warming ocean (He et al., 2023).

At intra-annual frequencies (M1 to M8-M16), the carbon flux anomalies are generally smaller but comparable to the "background" fluxes, and they exhibit greater variability at higher frequencies, indicating rapid events of carbon uptake and release. However, a pattern emerges: anticyclones tend to absorb more carbon or release less carbon than the "background", while cyclones typically absorb less or release more carbon. This pattern is the most pronounced in the Agulhas region, followed by the Brazil-Malvinas Confluence, and is much weaker in the Tasmania region. The contrast between cyclones and anticyclones is particularly distinguishable in the M1-M4 and M4-M8 bands. These frequency bands are linked to seasonal differences or

rapid events such as phytoplankton blooms (Fig. 4d-f). Keppler et al. (2024) identified a clear seasonal pattern, with cyclonic eddies in the ACC exhibiting outgassing in fall and carbon uptake in spring, while anticylonic eddies north of the ACC showed enhanced uptake during spring.

## 3.4 DIC as the key regulator of oceanic pCO<sub>2</sub> and its main role on CO<sub>2</sub> flux





The CO<sub>2</sub> flux is influenced by three components: (1) solubility factor  $(S_c)$ , driven by temperature and salinity, (2) piston velocity  $(k_w)$ , driven by wind speed and temperature, and (3) the air-sea pCO<sub>2</sub> difference  $(\Delta pCO_2 = pCO_2^{ocean} - pCO_2^{atm})$ . Appendix A details how each term  $(S_c, k_w)$ , and  $\Delta pCO_2$  is isolated for the analysis.

Among these factors,  $k_w$  and  $\Delta pCO_2$  are the dominant contributors to the total  $CO_2$  flux (Fig. 4a). In the Agulhas Retroflection and Brazil-Malvinas Confluence regions, the influence of  $\Delta pCO_2$  is larger in cyclonic eddies, periphery, and "background". However, in anticyclonic eddies,  $k_w$  has a stronger impact. This may result from reduced variability in surface  $\Delta pCO_2$ , as anticyclones contain warmer, lower-nutrient water, which limits biological activity and reduces  $pCO_2$  disequilibrium with the atmosphere. By contrast, in the Tasmania region  $\Delta pCO_2$  consistently dominates across all flow regimes. The contribution of  $S_c$  remains minimal across all regions and flow regimes, ranging from 10-13% (Fig. 4a).

The time-frequency analysis of the effect of the  $CO_2$  flux terms reveals a shift in the relative importance of these drivers. At higher frequencies,  $k_w$  becomes the dominant factor influencing  $CO_2$  flux, while at lower frequencies, the influence of  $\Delta pCO_2$  increases (Fig. 4b). Despite these variations,  $\Delta pCO_2$  remains a key driver of the total flux. (Fig. 4a). Our results align with those of Gu et al. (2023), which reported that, at higher frequencies (subseasonal timescales), wind explains the majority of the global  $CO_2$  flux anomaly, while at lower frequencies (seasonal, interannual and decadal timescales)  $\Delta pCO_2$  is the dominant factor. Our study expands on the factors that control the  $CO_2$  flux in eddies, showing that  $k_w$  tends to play a greater or comparable role in eddies than in the "background" regime.

In our simulation, the  $pCO_2^{atm}$  is prescribed, thus variations in  $\Delta pCO_2$  are primarily driven by changes in  $pCO_2^{ocean}$ . The  $pCO_2^{ocean}$  is driven by changes in DIC, alkalinity, sea surface salinity, and SST. To decompose the contributions of these factors, we applied a decomposition method (Takahashi et al., 1993), detailed in Appendix B. In the three selected regions, DIC and SST were the primary drivers of  $\Delta pCO_2^{ocean}$ . The contributions of alkalinity and salinity were relatively small; therefore, salinity was grouped with SST to represent the physical drivers, and alkalinity was grouped with DIC to represent the biogeochemical drivers. Because the salinity contribution was minor, the combined SST and salinity component is referred to as the thermal component, while the DIC and alkalinity component is referred to as the non-thermal component.

The regression of the  $pCO_2^{ocean}$  on the non-thermal component, primarily driven by DIC, exhibits high  $R^2$  values (Fig. 5), indicating that a larger proportion of  $pCO_2^{ocean}$  variability is explained by this component. Its dominance is present in the total signal as well across almost all frequency bands (except M4-M8) and regions. In contrast, the thermal component shows consistently low  $R^2$  values (Fig. 5a, b), suggesting a less relevant role in explaining the variability. It is worth mentioning that the thermal component counteracts the influence of the non-thermal component.

Other studies have found that DIC is the dominant driver of oceanic  $\Delta pCO_2$  in the Southern Ocean (Landschützer et al., 2016; Lerner et al., 2021). Our results reinforce this finding and further show that the drivers of oceanic  $pCO_2$  in eddies are

largely consistent with those in the surrounding "background" waters. Our results show that DIC is the dominant contributor to the non-thermal component of the  $pCO_2^{ocean}$ , which in turn exerts the strongest influence on air-sea  $CO_2$  fluxes. However, Smith et al. (2023) reported contrasting behavior in the South-East Atlantic Ocean, where temperature was identified as the main driver of  $pCO_2$  in eddies. Consequently, they observed enhanced  $CO_2$  uptake in cyclonic eddies compared to anticyclonic eddies. These findings highlight the spatial heterogeneity of eddy-related processes across the Southern Ocean and underscore the importance of regional processes in modulating air-sea  $CO_2$  fluxes.

Figure 5. Contribution of the thermal and non-thermal components to the total oceanic  $\Delta pCO_2$  across flow regimes in the defined regions. The coefficient of determination  $(R^2)$  was calculated from linear regressions between each component (thermal and non-thermal) and the total  $\Delta pCO_2$ , quantifying the proportion of variability explained by each. Regressions were performed separately for each component using time series of the composites within each region.

#### 4 Summary and conclusion



The purpose of our analysis was twofold: to characterize the role of mesoscale oceanographic features in modulating CO<sub>2</sub> fluxes across three eddy-rich regions of the Southern Ocean, and to investigate the physical and biogeochemical drivers that control these fluxes. We present a 27-year long high-resolution simulation, which is notably long for an ocean-biogeochemical model of this kind. Our simulation enables, for the first time, to assess the influence of eddies on CO<sub>2</sub> fluxes across a broad range of timescales, from daily to decadal variability. The model successfully captures mesoscale activity, reproducing the distinct characteristics of eddies: anticyclones associated with higher temperatures and lower DIC concentration, while cyclones exhibit lower temperatures and higher DIC concentrations.

We find that eddies act as a persistent carbon sink at decadal scales. At shorter timescales, mean flux anomalies are smaller but exhibit higher variability, with anticyclonic and cyclonic eddies displaying distinct behaviors. Anticyclones typically enhance carbon uptake or suppress outgassing, whereas cyclones more often reduce uptake or promote outgassing. This opposing behavior leads to a partial compensation between positive and negative flux anomalies, particularly at the basin scale, where en-

hanced CO<sub>2</sub> uptake by anticyclonic eddies tends to be offset by reduced uptake within cyclonic structures. Such compensation dampens the net contribution of mesoscale activity to the overall Southern Ocean carbon sink. These highlight the importance of eddy type in shaping short-term air-sea CO<sub>2</sub> fluxes.

Beyond these temporal dynamics, our results show that the mesoscale flow regimes (anticyclones, cyclones, and periphery) are more effective in taking up carbon than the "background", accounting for about 10% of the total carbon uptake in the Southern Ocean and 1% of the anomalous contribution. The relatively small overall value primarily reflects the large spatial extent of the domain, where "background" conditions dominate and contribute substantially to the total carbon uptake. However, in eddy-rich regions, the mesoscale contribution becomes much more important. Anticyclonic eddies are especially effective, emerging as the flow regime with the highest carbon uptake per unit area. These results are relevant given current trends suggesting that eddies may become more widespread in the future in the Southern Ocean.




Across all timescales and regions,  $\Delta pCO_2$  plays a central role in modulating the  $CO_2$  flux, outweighing the influence of piston velocity (wind) and solubility factor. We identified the non-thermal component (primarily driven by DIC) as the dominant control on oceanic  $\Delta pCO_2$ . This finding is consistent with previous studies on the Southern Ocean but extends our understanding by explicitly identifying the drivers of  $CO_2$  fluxes within eddies. Our results indicate that the factors regulating oceanic  $\Delta pCO_2$  in eddies are largely consistent with those in the surrounding "background" waters. However, given the differences we observe in  $CO_2$  uptake between eddies and the "background", further analysis is needed to determine the processes driving DIC variability within eddies and their impact on carbon fluxes.

The pattern of anticyclonic and cyclonic eddies in modulating  $CO_2$  fluxes in the Southern Ocean in our simulation is consistent with recent observational estimates. This confirms that the model realistically represents the influence of mesoscale eddies on  $CO_2$  fluxes. This agreement supports the model's ability to capture key physical and biogeochemical processes driving carbon uptake in the Southern Ocean. Moreover, we found that while the overall contribution of mesoscale eddies to carbon uptake across the entire Southern Ocean is relatively small, their influence becomes more pronounced in eddy-rich regions, leaving a detectable imprint on regional  $CO_2$  fluxes.

Code availability. Python scripts to reproduce the analysis are available at: https://doi.org/10.17617/3.LQT15X

Data availability. The data will be made publicly available at https://www.wdc-climate.de/ui/ at the latest when the manuscript is published.

Author contributions. MSM, NS and TI designed the study and led the analysis. FC, TI and members of TI's group developed the strategy for running the model. FC and MSM carried out the model simulations. MSM wrote the manuscript, with all authors contributing to its review and editing.

Competing interests. The authors declare that they have no competing interests.

Acknowledgements. MSM was supported by the German Research Foundation (DFG) under Germany's Excellence Strategy—EXC 2037 
"CLICCS—Climate, Climatic Change, and Society"—Project Number 390683824. TI and FC were supported by the EU Horizon 2020 
research and innovation program under grant agreement 101003536 (project ESM2025–Earth System Models for the Future). Computational 
resources were provided by the German Climate Computing Center (DKRZ). We thank Lucas Casaroli for providing the initial conditions 
used to initialize the 10 km resolution simulation. We are grateful to David Nielsen for his internal review, which helped improve the 
manuscript. We also thank Jacqueline Behncke for her valuable comments and feedback on the manuscript.

## Appendix A: air-sea CO2 flux terms effect

The air-sea  $CO_2$  flux follows the empirical relationship from Wanninkhof (1992). This is determined by the solubility factor  $(S_{CO_2})$ , the piston velocity  $(k_w)$ , and the air-sea  $pCO_2$  difference  $(\Delta pCO_2 = pCO_2^{ocean} - pCO_2^{atm})$ .

$$CO_{2flux} = S_{CO_2} k_w \Delta pCO_2$$

where  $k_w$  scales with wind speed and decays with temperature (Wanninkhof, 2014). The Schmidt number ( $S_c$ ) is fitted by the a 4th-order polynomial from Wanninkhof (2014) and normalized by the Schmidt number for  $CO_2$  at 20°C of 660:

$$k_w = (1-f) \ u^2 \Big(\frac{S_c(SST)}{660}\Big)^{-1/2}$$
 
$$S_c = a_1 - a_2 * SST + a_3 * SST^2 - a_4 * SST^3 + a_5 * SST^4$$
 
$$a_1 = 2116.8, \ a_2 = 136.25, \ a_3 = 4.7353, \ a_4 = 0.092307, \ a_5 = 0.0007555$$

and  $S_{CO_2}$  is calculated according to Weiss (1974)

$$lnS_{CO_2} = A_1 + A_2(100*SST) + A_3ln(SST/100) + SSS[B_1 + B_2(SST/100) + B_3(SST/100)^2]$$
 
$$A_1 = -60.2409, \ A_2 = 9345.17, \ A_3 = 23.3585, \ B_1 = 0.023517, \ B_2 = -0.00023656, \ B_3 = 0.004703656$$

where SST is sea surface temperature and SSS is sea surface salinity.

To isolate the effects of each term, we used the long-term mean state of the system as a baseline. By keeping the base-370 line constant for each driver, we could subtract its effect from the total CO<sub>2</sub> flux to determine the individual contributions (Chikamoto and DiNezio, 2021).

$$CO_{2flux} \approx F_{piston} + F_{solubility} + F_{\Delta pCO_2}$$

$$F_{piston} = k_w(t) \ S_{CO_2}(t) \ \Delta pCO_2(t) - \overline{k_w(t_b)} \ S_{CO_2}(t) \ \Delta pCO_2(t)$$

$$F_{solubility} = k_w(t) \ S_{CO_2}(t) \ \Delta pCO_2(t) - k_w(t) \ \overline{S_{CO_2}(t_b)} \ \Delta pCO_2(t)$$
$$F_{\Delta pCO_2} = k_w(t) \ S_{CO_2}(t) \ \Delta pCO_2(t) - k_w(t) \ S_{CO_2}(t) \ \overline{\Delta pCO_2(t_b)}$$

where  $\overline{k_w(t_b)}$ ,  $\overline{S_{CO_2}(t_b)}$ , and  $\overline{\Delta pCO_2(t_b)}$  are the respective baseline values, defined as their long-term means.

The combined effects of each driver should approximate the total  $CO_2$  flux. The reconstruction is an approximation because it does not account for the effects of sea ice, although our analysis was restricted to sea-ice-free regions.

# 380 Appendix B: Decomposition of $pCO_2^{ocean}$

Since  $pCO_2^{air}$  is prescribed in the simulation, the primary variations in  $\Delta pCO_2$  are driven by changes in  $pCO_2^{ocean}$ . The  $pCO_2^{ocean}$  is influenced by variations in DIC, alkalinity, sea surface salinity, and temperature. To decompose the  $pCO_2^{ocean}$ , we applied a widely used decomposition method (Takahashi et al., 1993).

$$\Delta pCO_2^{ocean} = \Delta pCO_2^{DIC} + \Delta pCO_2^{alk} + \Delta pCO_2^{SST} + \Delta pCO_2^{SSS}$$

$$\Delta pCO_2^{DIC} = \Delta DIC * \gamma_{DIC} * pCO_{2Ref} / \overline{DIC}$$

$$\Delta pCO_2^{alk} = \Delta alk * \gamma_{alk} * pCO_{2Ref} / \overline{alk}$$

$$\Delta pCO_2^{SSS} = \Delta SSS * \gamma_{SSS} * pCO_{2Ref} / \overline{SSS}$$

$$\Delta pCO_2^{SST} = (\exp^{\Delta SST * \gamma_{SST}} * pCO_{2Ref}) - pCO_{2Ref}$$

The values of  $\gamma_X$  refer to the Revelle factors.  $\gamma_{SSS}$  and  $\gamma_{SST}$  are 1 and 0.0423;  $\gamma_{DIC}$  and  $\gamma_{alk}$  are computed daily as (Sarmiento and Gruber, 2006):

$$\begin{split} \gamma_{DIC} &= \frac{3 \ alk \ DIC - 2 \ DIC^2}{\left(2 \ DIC - alk\right) \left(alk - DIC\right)} \\ \gamma_{alk} &= -\frac{alk^2}{\left(2 \ DIC - alk\right) \left(alk - DIC\right)} \end{split}$$

#### 395 Appendix C: Selection of the OW threshold

Figure C1 shows the OW fields for different threshold values. This comparison demonstrates that the selected threshold provides the most adequate balance between capturing coherent eddy cores and avoiding overly extended periphery regions.

## Okubo-Weiss parameter $[s^{-2}]$ 2014-06-03

Figure C1. Sensitivity of mesoscale structure detection to different OW threshold values. The panels show four threshold selections used to define mesoscale regimes. The threshold of  $0.3(\sigma_{OW})$  was used in this study.

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
