# Peer review of "Mesoscale eddies heterogeneously modulate CO2 fluxes in eddy-rich regions of the Southern Ocean"

_EGUsphere, 2025_

## Author Comment (AC1)

Salinas-Matus et al. present an investigation into the effects of mesoscale eddies on the air-sea CO2 flux within eddy rich regions of the Southern Ocean. They find geographical differences in the eddy effects on air-sea CO2 flux between the Brazil-Malvinas confluence, Agulhas Retroflection and the region South of Tasmania. They investigate the factors driving these differences in the air-sea CO2 fluxes and find that the pCO2 gradient ( $\Delta$ pCO2) is the primary driver, which itself is mainly driven by differences in dissolved inorganic carbon (DIC). Using a temporal decomposition a further result shows that eddies act as persistent CO2 sinks on decadal timescales but are more variable at shorter timescales. I find this study to provide a comprehensive analysis of the eddy rich regions of the Southern Ocean and the modulation of the air-sea CO2 fluxes and can recommend for publication once my remaining comments below have been addressed (especially the analysis in Figure 5 which is unclear).

We would like to sincerely thank the reviewer for the very helpful and constructive comments and suggestions. We have carefully addressed each point below. The reviewer's comments are shown in bold font, and our responses are provided in normal font.

General comment: I'd suggest to aid readers, to change the cyclonic eddy results to be coloured blue, and the anticyclonic eddies to be coloured red. Although the legends are clear in defining the colours, the common conventions in previous work are for anticyclonic (generally 'warm') eddies are coloured red, and cyclonic (generally 'cold') eddies blue.

Yes, thanks for the suggestion. As you said, it's better to keep the convention that has already been used in other papers. In the new version, the colours have been reverted.

General comment: I'd suggest picking a different colour for the periphery to avoid having red and green to aid readers that are colour-blind.

Yes, completely makes sense. Done

Line 27: "Cyclone" should be "Cyclonic" Done

Line 86: Can some details of the air-sea CO2 flux parameterisation be mentioned? What was the parameterisation used for kw (and Schmidt number), and the solubility.

Thank you for pointing this out. We have now added details in Appendix A describing how the piston velocity (kw), Schmidt number, and solubility factor were computed for the analysis of the contribution of each term. These parameterisations are the same as those implemented in the model itself, which is why we reconstruct the CO2 flux consistently with the model output. Specifically, we use Wanninkhof (2014) for both the Schmidt number and the gas transfer velocity, and Weiss (1974) for the solubility factor. The corresponding references have been added to the manuscript.

**Appendix A: air-sea CO2 flux terms effect**

The air-sea  $CO_2$  flux follows the empirical relationship from Wanninkhof (1992). This is determined by the solubility factor  $(S_{CO_2})$ , the piston velocity  $(k_w)$ , and the air-sea  $pCO_2$  difference  $(\Delta pCO_2 = pCO_2^{ocean} - pCO_2^{atm})$ .

$$CO_{2flux} = S_{CO_2} k_w \Delta p CO_2$$

where  $k_w$  scales with wind speed and decays with temperature (Wanninkhof, 2014). The Schmidt number ( $S_c$ ) is fitted by the a 4th-order polynomial from Wanninkhof (2014) and normalized by the Schmidt number for  $CO_2$  at 20°C of 660:

$$k_w = (1-f) \; u^2 \Big(\frac{S_c(SST)}{660}\Big)^{-1/2}$$

$$S_c = a_1 - a_2 * SST + a_3 * SST^2 - a_4 * SST^3 + a_5 * SST^4$$

$$a_1 = 2116.8, \; a_2 = 136.25, \; a_3 = 4.7353, \; a_4 = 0.092307, \; a_5 = 0.0007555$$

and  $S_{CO_2}$  is calculated according to Weiss (1974)

$$\begin{split} lnS_{CO_2} &= A_1 + A_2(100*SST) + A_3ln(SST/100) + SSS[B_1 + B_2(SST/100) + B_3(SST/100)^2] \\ A_1 &= -60.2409, \ A_2 = 9345.17, \ A_3 = 23.3585, \ B_1 = 0.023517, \ B_2 = -0.00023656, \ B_3 = 0.0047036 \end{split}$$

where SST is sea surface temperature and SSS is sea surface salinity.

To isolate the effects of each term, we used the long-term mean state of the system as a baseline. By keeping the base-360 line constant for each driver, we could subtract its effect from the total CO2 flux to determine the individual contributions (Chikamoto and DiNezio, 2021).

$$CO_{2flux} \approx F_{piston} + F_{solubility} + F_{\Delta pCO_2}$$

$$F_{piston} = k_w(t) \ S_{CO_2}(t) \ \Delta pCO_2(t) - \overline{k_w(t_b)} \ S_{CO_2}(t) \ \Delta pCO_2(t)$$

$$F_{solubility} = k_w(t) \ S_{CO_2}(t) \ \Delta pCO_2(t) - k_w(t) \ \overline{S_{CO_2}(t_b)} \ \Delta pCO_2(t)$$

$$F_{\Delta pCO_2} = k_w(t) \ S_{CO_2}(t) \ \Delta pCO_2(t) - k_w(t) \ S_{CO_2}(t) \ \overline{\Delta pCO_2(t_b)}$$

where  $\overline{k_w(t_b)}$ ,  $\overline{S_{CO_2}(t_b)}$ , and  $\overline{\Delta pCO_2(t_b)}$  are the respective baseline values, defined as their long-term means.

The combined effects of each driver should approximate the total  $CO_2$  flux. The reconstruction is an approximation because it does not account for the effects of sea ice, although our analysis was restricted to sea-ice-free regions.

**Line 92: What was the native time resolution of the model, that was then collated into the daily averages?**

The time step of the model is 600 seconds, and as you mentioned, then a daily average is done.

Line 113: What is the sensitivity of the results to the Okubo-Weiss parameter values used to define the background, eddy and periphery. Is this a commonly applied value (I don't see a reference for this selection)? The selection appears to identify large regions far from negative OW values, that get identified as the "periphery" in Figure 1a, which may suggest that this value is too relaxed?

Thank you for this important comment. We have expanded in the Eddy detection method section a clarification of the procedure used to determine the Okubo–Weiss (OW) threshold and provided references to previous studies. The text added is:

"...The "background" regime is defined using a threshold equal to 0.3 of the temporal mean of the spatial standard deviation of the OW ( $\sigma_{OW}$ ) corresponding to  $\pm 0.5 \times 10^{-10} s^{-2}$ , encompassing all OW values within this range. The most common approach in the literature is to define the threshold as  $0.2\sigma_{OW}$  (Schütte et al., 2016a; Vu et al., 2018), with some studies adopting more relaxed values around  $0.1\sigma_{OW}$  (Beech et al., 2025). However, in this study we chose to apply a slightly stricter criterion in order to isolate more robust and well-defined eddy and mesoscale structures, while still retaining the main mesoscale (see supplementary figure S1)."

We have added a supplementary figure showing the OW for different threshold values. This comparison demonstrates that the selected threshold provides the most adequate balance between capturing coherent eddy cores and avoiding overly extended "periphery" regions.

**Okubo-Weiss parameter $[s^{-2}]$ 2014-06-03**

Figure S1. Sensitivity of mesoscale structure detection to different Okubo–Weiss (OW) threshold values. The panels show four threshold selections used to define mesoscale regimes. The threshold of 0.3σOW was used in this study, as it provides a balance between capturing well-defined eddy structures and preserving the main mesoscale variability.

Additionally, a sensitivity test was performed for the periphery and background, evaluating threshold limits of 0.2σOW, 0.3σOW, and 0.4σOW. The results show same variability, and the differences in magnitude among these thresholds are minimal, confirming that the results are not sensitive to the specific choice of threshold.

Figure: Results of the sensitivity analysis evaluating the impact of different threshold selections used to define the periphery regime in the Agulhas Retroflection region. The figure illustrates how variations in the threshold affect the composites of air–sea CO2 fluxes.

**Line 130: I'd suggest the final sentence isn't required - it could be moved to the conclusion or introduction if the authors would like to keep it.**

We agree with you that the final sentence was not necessary in that section. Accordingly, we have removed it from the manuscript.

**Figure 1: Suggest changing colour map in Figure 2b (bottom panel) for colourblindness.**

Done, we changed it to another color palette that is friendly for people with color blindness.

Line 170: Id suggest more is said surrounding the "background" conditions in the Agulhas retroflection region being cooler than the cyclonic eddies. The cyclonic eddies would be forming from the cool side of the retroflection and therefore should be cooler than the background as this water wouldn't be originating in the Indian Ocean. Figure 3a shows the cooler water on Southern Ocean side of the retroflection, and Figure 2b shows SST anomalies for cyclonic eddies being negative. The background conditions also appear based on Figure 1a to be originating from the Indian South Subtropical gyre with warmer temperatures. We thank the reviewer for this careful comment, which we had initially overlooked. We identified labeling errors in Figure 3b, during the creation of the Python dictionary used to assign the data to each flow regime, the cyclonic composites were incorrectly mapped to the periphery keyword, the periphery to background, and the background to cyclonic. We have now corrected Figure 3b accordingly. In addition, we reviewed all other figures and confirmed that no similar labeling errors are present.

**We modified the text as follows:**

"In our simulations, flow regimes exhibit distinct characteristics that influence the CO2 flux

(Fig. 2b-d), with contrasting vertical structures of cyclonic and anticyclonic eddies emerging (Fig. 3b), consistent with previous findings (Keppler et al., 2024). Anticyclones, which tend to induce water downwelling, exhibit higher temperatures and lower DIC concentrations. In contrast, cyclones exhibit lower temperatures and higher DIC concentrations, mostly driven by water upwelling. The periphery regime shows intermediate properties, representing the transition zones surrounding both anticyclonic and cyclonic eddies (Fig. 3b). While this general pattern is evident across all three regions, the magnitude of the differences varies with latitude and local dynamics, reflecting the influence of regional circulation."

Figure 3. ... (b) Vertical temperature and DIC profile composites for the flow regimes within the defined regions, ...

Line 201: Id suggest more is said on the Tasmania region eddies and the sporadic events. These sporadic high uptake events seem to be more prevalent in the Tasmania region, compared to the other regions, and have a large effect on the Figure 4 uptake results. As pCO2 is the dominant driver in the integrated fluxes, could you elaborate on a mechanism?

The paragraph as previously written could be misleading. What we meant is that strong carbon uptake events are present across all flow regimes, including the background, and that when integrating over all regimes, the net result for the region is  $CO_2$  uptake. We have also added a reference discussing periods of enhanced upwelling that transport DIC from the deeper ocean, thereby reducing carbon uptake in the region. In addition, we expanded the discussion on the anomalous contribution of mesoscale regimes, following the suggestion from Reviewer 2. The revised paragraph now reads:

"Tasmania region exhibits a contrasting pattern, as the "background" has the highest CO2 uptake (Fig. 4a). Here, both anticyclonic and cyclonic eddies take up less or outgas more carbon (Fig. 4c). This pattern is influenced by the weaker eddy intensity (Fig. 1c) and the characteristics of the region. Most eddies in the Tasmania region are located in the high-DIC band, which limits the ocean's capacity for CO2 uptake (Fig. 3b). Nevertheless, anticyclonic and cyclonic eddies take up carbon in the entire analyzed period (Fig. 4a), a pattern influenced by sporadic but intense carbon uptake events occurring across all flow regimes in the region (Fig. 3). These events may be associated with periods of reduced upwelling activity, which limit the vertical transport of DIC-rich subsurface waters to the surface, thereby allowing enhanced CO2 uptake (Pardo et al., 2017). However, since the efficiency of CO2 uptake in mesoscale flow regimes is lower than in the background, the anomalous contribution is negative. Mesoscale regimes take up approximately 17% less carbon compared to what would be expected under background conditions. (Table 3)."

**Table 2: It is unclear what the $\pm$ values denote. Is it mean $\pm$ the standard deviation? We added a clarification in the table description:**

"Table 2. Area-integrated carbon uptake contributions by region and mesoscale flow regime. The value beneath each region name indicates the total carbon uptake (including background, eddies, and periphery). The values in parentheses represent the contribution of each flow regime as a percentage, first relative to the total uptake (in bold), and second relative to the combined uptake from all mesoscale flow regimes (eddies + periphery). All values are reported as mean ± standard deviation."

**Line 253: Could the greater influence of kw be due to the increase in wind speeds generally observed over anticyclonic eddies? (Frenger et al., 2013) and the opposite for cyclonic eddies?**

Indeed, some observational and modelling studies have shown that mesoscale SST anomalies associated with eddies can imprint on the overlying atmosphere, producing local wind-speed anomalies that tend to be positive over warm (anticyclonic) anomalies and negative over cold (cyclonic) anomalies (e.g. Frenger et al., 2013; He et al., 2020; Ji et al., 2020). However, our study uses an ocean-only model forced by prescribed atmospheric fields (two-way atmosphere-ocean is absent). Therefore, eddy-scale modifications of wind speed are only present in our simulations if they already exist in the prescribed atmospheric forcing; they are not generated dynamically by the model. In our case, the atmospheric forcing is ERA5 (as stated in the Experiment Design section). ERA5 does contain some representation of eddy-scale wind anomalies, although the signal is generally attenuated compared to satellite-based products or fully coupled simulations. This constitutes a limitation of our study, We have added a sentence in the Experiment Design section to clarify this:

"... As this is an ocean-only configuration without coupling to the atmosphere, eddy-induced feedbacks on surface winds are not represented."

**Line 264: How was the pCO2 (atm) prescribed? Can details of this be added in the methods for model setup?**

The atmospheric pCO2 used in the model was prescribed from the Global CO2 concentration dataset (in ppm) prepared for the Global Carbon budget (Friedlingstein et

al., 2022). We have now added the citation in the Experiment design section: "...The atmospheric pCO2 used in the model was prescribed from the global CO2 concentration dataset prepared by the Global Carbon Project (Friedlingstein et al., 2022)."

Line 271: Figure 5 captions indicate these regressions are for ΔpCO2 regressed against the thermal and non-thermal components separately, where as the text indicates this is the total flux. Based on the regressions I think this is each component regressed against the total flux. The analysis in this form may not answer the aim as the contributions of the thermal and non-thermal components to the changes in ΔpCO2 would be combined with variability in kw (and other inputs to the fluxes) when regressing against fluxes. I am unsure of the aim of this portion of the analysis and suggest the authors should clarify this analysis (Lines 264-275). Thank you very much for pointing this out. The text is incorrectly phrased. What we actually did in this analysis was to regress oceanic  $\Delta pCO2$  against its thermal and nonthermal components separately, as correctly shown in the description of Figure 5. The mention of the "total flux" in the text was a writing error that escaped our revision. We are not sure why this may have given the impression that the regression was against the total flux, but we will clarify the wording to avoid any possible confusion. What we had was the time series of ΔpCO2 decomposed into its thermal and non-thermal components, which add up to the total  $\Delta pCO2$ . Each of these components was regressed against  $\Delta pCO2$  in order to explain the variability.

We corrected the text accordingly to clarify that the regressions refer to  $\Delta$ pCO2 and its components, not to the total flux:

"The regression of the pCO2ocean on the non-thermal component, primarily driven by DIC, exhibits high R2 values (Fig. 5), indicating that a larger proportion of pCO2ocean variability is explained by this component."

Code availability: Code is available to do sections of the analysis but was unable to find plotting scripts to complete the analysis.

The scripts used to generate the figures will be included in the same repository as the analysis code when the revised version of the manuscript is resubmitted.

Data availability: I note no data availability statement. Are these fields available or can they be requested? I expect not due to data volume, but this should be stated. These fields appear very useful for studying mesoscale eddies over a long time period, so could be useful for the community.

A data availability statement will be included in the revised submission. While the full dataset is very large, we will provide clear information on how the fields can be accessed or requested.

**References**

Frenger, I., Gruber, N., Knutti, R., & Münnich, M. (2013). Imprint of Southern Ocean eddies on winds, clouds and rainfall. Nature Geoscience, 6(8), 608–612. https://doi.org/10.1038/ngeo1863

---

## Author Comment (AC2)

Salinas-Matus et al. investigated the influence of anticyclonic and cyclonic eddies, periphery and "background" in modulating air-sea CO2 fluxes. For this, three different regions with differing characteristics in the Southern Ocean were picked, namely the Agulhas Region, the Brazil-Malvinas Confluence Region and the region south of Tasmania. The aim of their study is to investigate the physical and biogeochemical drivers of the air-sea fluxes at daily to decadal timescales. Mesoscale eddies in the Southern Ocean play a key role in shaping air-sea CO2 fluxes and carbon uptake. Using a 27-year, global eddy-rich (nominal resolution 8.4-10 km) ocean biogeochemical simulation (ICON-O with HAMOCC), eddies were identified via the Okubo-Weiss parameter and analyzed across their cores, peripheries, and surrounding waters ("background"). Since the eddies were not tracked, the lifetime of the eddies was not considered in their study. The study highlights that the influence of eddies on CO2 fluxes is strongly regiondependent, with controlling mechanisms largely consistent with those in background waters. Variations in air-sea CO2 exchange are primarily driven by changes in the ocean-atmosphere pCO2 disequilibrium ( $\Delta$ pCO2), itself dominated by dissolved inorganic carbon (DIC) dynamics.

Anticyclonic eddies consistently show the strongest ingassing per unit area, which they attribute to eddy-pumping processes that transfer carbon to deeper layers, thereby enhancing long-term uptake capacity. Importantly, eddy peripheries emerge as a major contributor to regional CO2 fluxes: while their uptake efficiency is intermediate between eddy cores and background waters, their much larger spatial coverage makes them an integral component of the Southern Ocean carbon budget which requires further analysis.

On decadal timescales, mesoscale eddies emerge as a net carbon sink, contributing roughly 10% of the Southern Ocean's total carbon uptake if integrated over the area that the eddies cover. On shorter timescales, however, flux anomalies are more variable and reflect contrasting behaviors of cyclones and anticyclones.

We would like to thank the reviewer for the thorough evaluation of our manuscript and for the insightful and constructive comments provided. We appreciate the time and effort invested in reading our work and offering valuable suggestions that have helped us to improve the quality and clarity of the paper. The reviewer's comments are shown in bold font, and our responses are provided in normal font below.

**General comments:**

The article reads very smoothly and is easily understandable. The Abstract and Introduction highlight the relevance of the research interest which is also justified by profound literature references. General information about the model and the experiment design which is crucial in order to evaluate the results is given. The presentation of the results is in a logical order, easy to follow with the presented Figures and sufficiently substantiated with observational literature. The summary rounds up the paper in a very nice way, clearing all questions presented in the introduction and pointing out the main findings. Overall, Salinas-Matus et al. is an enjoyably written paper with interesting new insights on the driving mechanisms of eddies on  $CO_2$  fluxes that may need some clarification in some methodological details as well as a more detailed discussion about the overall impact of eddies on the Southern Ocean  $CO_2$  fluxes (see comments below) .

We sincerely appreciate the reviewer's comments regarding the clarity, logical flow, and relevance of our study. We would like to acknowledge that we initially overlooked citing a relevant study. To address this, we have now incorporated a reference to *Smith et al.* (2023) and revised the introduction to include this work, thereby providing better context for our study. The updated paragraph now reads as follows:

"... From a modeling perspective, Guo and Timmermans (2024) used an eddy-resolving global biogeochemical simulation to isolate the effect of mesoscale activity. Their findings indicate that mesoscale dynamics account for over 30 % of the total variance of the airsea CO2 fluxes in energetic regions and can act either as sources or sinks depending on the region, showing the importance of mesoscale dynamics from a global perspective. Smith et al. (2023) analyzed the heat and carbon characteristics of mesoscale eddies in the South–East Atlantic Ocean using a regional model configuration, revealing distinct thermodynamic and biogeochemical signatures between cyclonic and anticyclonic eddies. However, no modeling study to date has investigated the contrasting impacts of cyclonic and anticyclonic eddies on air–sea CO2 fluxes across the entire Southern Ocean, while also considering regional differences."

**Specific comments:**

L115ff: Is there literature based on which the threshold values were defined?
 (Common definition e.g. 0.2\* stdev(OW) as Schütte et al. 2016)

We have incorporated some references and clarified the literature basis for the definition of the threshold. In the revised manuscript, we have added the following explanation:

"...The "background" regime is defined using a threshold equal to 0.3 of the temporal mean of the spatial standard deviation of the OW ( $\sigma_{OW}$ ) corresponding to  $\pm 0.5 \times 10-10s^{-2}$ , encompassing all OW values within this range. The most common approach in the literature is to define the threshold as  $0.2\sigma_{OW}$  (Schütte et al., 2016a; Vu et al., 2018), with some studies adopting more relaxed values around  $0.1\sigma_{OW}$  (Beech et al., 2025). However, in this study we chose to apply a slightly stricter criterion in order to isolate more robust and well-defined eddy and mesoscale structures, while still retaining the main mesoscale (see supplementary figure S1)."

Figure S1. Sensitivity of mesoscale structure detection to different Okubo–Weiss (OW) threshold values. The panels show four threshold selections used to define mesoscale regimes. The threshold of  $0.3\sigma_{\text{OW}}$  was used in this study, as it provides a balance between capturing well-defined eddy structures and preserving the main mesoscale variability.

**Fig.2.: Some eddies look very deformed and unusually large → did the authors check for several local maxima/minima?**

We did not explicitly check for multiple local maxima/minima within each identified eddy because the Okubo–Weiss parameter defines eddy boundaries based on the balance between relative vorticity and strain, rather than on local extrema. With this methodology, you're right, in some cases, especially where two cyclonic or two anticyclonic structures are located very close to each other, the methodology may capture them as a single, elongated contour. Additionally, eddies near strong shear zones can also appear deformed due to the local flow field. However, in our composite analyses, the results are based on the total area encompassed by the detected eddy structures. Therefore, merging two nearby eddies into a single contour would not affect the aggregated statistics or the fluxes reported. We acknowledge, nevertheless, that this represents a methodological limitation inherent to the Okubo–Weiss–based eddy detection approach.

L126: could you elaborate on how you preprocess the data prior carrying out the Fourier analysis? The eddies are at different locations and several for each timestep. Is the average of "eddy composites" (defined as OW below the chosen threshold) taken for each time step (daily) and then the Fourier analysis on the such obtained time series applied? If this is the case, maybe a different term than "composite" should be used or at least defined.

Thank you for the opportunity to clarify the methodology. The Fourier analysis was not applied to the eddy composites. Instead, the spectral decomposition was performed first

on the full spatial variable fields. Specifically, we applied a band-pass Fourier filtering to the complete dataset, obtaining five filtered fields corresponding to the defined frequency bands. Subsequently, eddy composites were computed from each of these filtered fields, using the same thresholding procedure described earlier. This sequence is methodologically appropriate, as it preserves the full spatial and temporal coherence prior to compositing. In the revised manuscript, we have added the following:

"The net CO2 flux is influenced by processes operating across different timescales (Gu et al., 2023). To determine the frequency range at which mesoscale eddies have the greatest impact on CO2 flux, we first applied a Fourier analysis to the full spatial variable fields to decompose it into distinct frequency bands. This approach ensures that the spatial and temporal coherence of the field is preserved before the identification of eddies. The analysis yielded five filtered fields corresponding to (1) intra-annual variations above 16 months and up to 27 years (M16), (2) annual variations between 8 and 16 months (M8-M16), (3) intra-annual variations between 4 and 8 months (M4–M8), (4) intra-annual variations between 1 and 4 months (M1–M4), and (5) submonthly variations shorter than 1 month, down to daily changes (M1). Subsequently, eddy composites were computed for each filtered field using the criterion described above."

**• E.g., Fig. 2: How are anomalies defined? Also, tell how is the sign of the flux defined wrt ingassing/outgassing?**

The anomalies were calculated by subtracting the long-term mean from each variable at every grid point, i.e., anomaly = variable – long-term mean(variable). The air–sea CO2 flux is positive for outgassing and negative for uptake; therefore a positive flux anomaly corresponds to an increase in outgassing relative to the mean while a negative anomaly corresponds to an increase in uptake relative to the mean. In the revised manuscript, we have added the following in the figure description:

"Figure 2. Snapshot of the identified mesoscale dynamics and tracer or tracer flux anomalies in the Agulhas Retroflection region (defined in Fig. 1b) on 03 June 2014. (a) Okubo–Weiss parameter, delineating three flow regimes: vorticity-dominated (green), deformation-dominated (violet), and "background" (grey). (b) Sea Surface Temperature (SST) anomaly, (c) air–sea CO2 flux anomaly, and (d) Surface DIC anomaly. Anomalies represent deviations from the climatological state, computed as the difference between the instantaneous value and the long-term mean at each grid point. The air–sea CO2 flux is defined as positive for outgassing (flux from the ocean to the atmosphere) and negative for uptake (flux from the atmosphere to the ocean). Therefore, a positive flux anomaly indicates increased outgassing relative to the mean, while a negative anomaly indicates increased uptake relative to the mean. In all panels, contours outline anticyclonic and cyclonic eddies."

Eddy contribution to the Southern Ocean carbon sink: the authors provide % numbers for the contribution of eddies given as flux integrated over the "eddy" (= OW below threshold) area. How much bigger is this flux than if one assumed "background flux conditions" over the same area (in %)?

We agree with the reviewer that reporting the anomalous contribution is indeed informative. Accordingly, we have added a new table to present the anomalous eddy contribution, expressed as the percentage difference relative to the flux that would be expected under background conditions over the same area. As a result, the overall

**percentage contribution of eddies and periphery to the total Southern Ocean carbon sink**

**Table 3.** Same as table 2, but the anomalous contributions relative to the background associated with mesoscale regime flows. In other words, this table quantifies how much additional (or reduced, when negative) carbon uptake occurs due to mesoscale activity, representing the difference in carbon uptake that would be observed if only background conditions were considered.

| Region                | Mesoscale flow regimes       | Disentangling Mesoscale flow regimes |                        |                        |
|-----------------------|------------------------------|--------------------------------------|------------------------|------------------------|
|                       |                              | Anticyclones                         | Cyclones               | Periphery              |
| Agulhas Retroflection | 0.01150 ± 0.009              | $0.00428 \pm 0.00030$                | $0.00056 \pm 0.00134$  | $0.00614 \pm 0.00507$  |
|                       | ( 10 ± 0.08% ) | $(37 \pm 2\%)$                       | (5 ± 12%)              | (53 ± 44%)             |
| Brazil-Malvinas       | 0.00348 ± 0.006              | $0.00091 \pm 0.00138$                | $0.00017 \pm 0.00080$  | $0.00240 \pm 0.00388$  |
| Confluence            | ( 4 ± 0.06% )  | (26 ± 39%)                           | $(5 \pm 23\%)$         | (69 ± 111%)            |
| South Tasmania        | $-0.00298 \pm 0.002$         | $-0.000591 \pm 0.00053$              | $-0.00074 \pm 0.00061$ | $-0.00165 \pm 0.00110$ |
|                       | (-17 $\pm$ 11%)              | $(-20 \pm 18\%)$                     | $(-25 \pm 20\%)$       | $(-55 \pm 36\%)$       |
| Entire SO             | $0.00946 \pm 0.027$          | $0.00313 \pm 0.00531$                | $-0.00136 \pm 0.00414$ | $0.00769 \pm 0.0180$   |
|                       | (1±2.5%)                     | (33 ± 56%)                           | $(-14 \pm 43\%)$       | (81 ± 190%)            |

decreases substantially. Here the new table in the revised manuscript: Nevertheless, we also kept the original table, since the two perspectives provide complementary information. While the anomalous contribution helps to quantify the relative enhancement or reduction induced by mesoscale activity, the total flux over the eddy-covered area remains relevant for assessing the integrated role of eddies. Moreover, it is important to note that the "what-if" scenario, assuming background flux conditions in the absence of eddies, represents a purely theoretical construct. In reality, the background state itself would likely be modified by the absence of mesoscale structures, as eddies strongly influence the mean circulation, stratification, and nutrient distributions. Therefore, the anomalous flux values should be interpreted as an idealized sensitivity test rather than a physically realizable alternative state.

**Along similar lines, the per square meter fluxes (Fig. 4c), how different are they compared to the background fluxes (in %)?**

We have addressed the reviewer's point. In the revised version of Figure 4c, we have added the mean background CO2 flux value as a reference line, which allows direct visual comparison of how much stronger or weaker the eddy-associated fluxes are relative to background conditions.

Figure 4. ... (c) Composite anomalies of total CO2 flux relative to the "background" for anticyclonic and cyclonic eddies, including the mean background value...

I am asking as such a statement (of 10-20% contribution by eddies to the Southern Ocean carbon sink) makes the eddy contribution sound very important. The anomalous contribution (i.e., "what if" the area covered by eddies was not covered by eddies) is more interesting. There is a lot of compensation between anticylones, cyclones, and perhaps seasonality, as the authors point out themselves. E.g., they cite Song et al., 2016, in the introduction (polarity dependence of fluxes), but do not get back to it in the discussion. How do the authors' findings compare?

The first part of this question regarding the anomalous contribution of eddies has already been addressed in above responses. The partial compensation between positive and negative CO2 flux anomalies is now explicitly discussed in both the Results and mentioned in the Conclusions sections:

"Despite regional heterogeneity, anticyclonic eddies take up more or outgas less carbon than the background, while cyclonic eddies exhibit the opposite pattern (Fig. 4c). This is consistent with studies using observation-based datasets (Keppler et al., 2024; Li et al., 2025). This contrasting behavior may be attributed to eddy-pumping. In anticyclonic eddies, this mechanism facilitates the transfer of carbon to deeper ocean layers, enhancing the ocean's capacity for carbon uptake over time. In contrast, cyclonic eddies may act to reduce carbon uptake by limiting this downward transport. While this opposition can lead to a partial compensation between positive and negative flux anomalies over the entire Southern Ocean, such compensation is not observed at the regional scale (Table 3)."

Regarding seasonality, we agree that interesting patterns likely exist. However, in this study we chose not to focus on seasonal variations, as this is currently the subject of ongoing work in which we aim to provide a detailed analysis of the mechanisms driving seasonal CO2 flux changes associated with mesoscale eddies.

• Given the above two points, I do not feel that comfortable with a strong statement that eddies are very important for the Southern Ocean carbon sink (because of the local air-sea flux anomalies they cause, which is the focus of the paper); are the authors? I would be more comfortable if they discussed the above aspects in their manuscript more clearly/transparently.

We agree with the reviewer that the previous wording in the conclusion section may have overstated the overall importance of eddies in the Southern Ocean carbon sink. In the revised manuscript, the percentages have been updated to reflect the anomalous contributions of eddies and periphery, which are notably smaller than the originally reported total fluxes. Accordingly, the concluding statement has been rephrased to show the role of mesoscale eddies as modulators of CO2 fluxes rather than as dominant contributors, which could have been implied by the previous version.

 L86 or L252ff: mention somewhere, that, given that this is an ocean-only model (no coupling to the atmosphere) there is no eddy feedback on winds, i.e., a bit of the effect eddies might have on air-sea CO2 fluxes is missing?

The Reviewer is correct. Since this is an ocean-only model, eddy-induced feedbacks on surface winds are not represented. We have added a sentence in the Experiment Design section to clarify this:

- "...As this is an ocean-only configuration without coupling to the atmosphere, eddy-induced feedbacks on surface winds are not represented."
  - L103: the authors discuss the drift of the coarser resolution model, can they
    also briefly comment on the (likely) drift of the 10 km model (which was run for
    a much shorter time period)

The drift mentioned refers to the 10 km high-resolution model (control simulation). We have clarified this in the revised manuscript:

- "... Furthermore, all model drifts in the 10 km resolution control run remain sufficiently small in accordance with the CMIP6 protocol (Jones et al., 2016)."
  - L151: Can the authors elaborate on how/why they expect the eddy intensity to relate to air-sea CO2 fluxes?

We added this to the sentence to be clear why eddy intensity could be related to the airsea co2 fluxes:

- "... For instance, the eddy intensity differs between regions and is expected to modulate air—sea CO2 fluxes, as stronger eddies enhance both lateral and vertical transport of water properties, including temperature and
- enhance both lateral and vertical transport of water properties, including temperature a DIC."
  - L189 "mesoscale flow regimes (anticyclones, cyclones, and periphery) have a
    greater capacity for carbon uptake compared to the "background" (Fig. 4a)":
    not true for all regions, and not for all of the SO?

The Reviewer is right, this statement does not apply uniformly across all regions of the Southern Ocean. We have modified the text to reflect that mesoscale regimes generally show higher efficiency than the background, but with regional exceptions. In the revised manuscript, we have added:

"The integrated CO2 flux over the 27 analyzed years indicates that, in most regions, mesoscale flow regimes (anticyclones, cyclones, and periphery) exhibit a greater efficiency in carbon uptake compared to the background, although this pattern is not consistent across the entire Southern Ocean (Fig. 4a)"

 L192 "Among these regimes, anticyclonic eddies show an enhanced ability to take up CO2 (Fig. 4a).": see comment above, how large is the enhanced ability, in absolute numbers and relative to the background flux?

We agree with the reviewer that specifying the relative magnitude of the enhanced CO2 uptake provides valuable context, and we have already modified this point as mentioned in the previous responses. In this case, the comparison can be directly inferred from Figure 4a, where the bars representing each flow regime are shown side by side. For the Agulhas Retroflection region, which this sentence refers to, the CO2 uptake in anticyclonic eddies is approximately twice that of the background flux. This relative difference can be readily appreciated in the figure, as the bar corresponding to anticyclones is roughly double the height of that for the background.

 L197 "and the stronger vertical gradient in those two regions (Fig. 3b).": do one clearly see this from the Figure?

Yes, the stronger vertical gradients in the Brazil–Malvinas Confluence and Agulhas regions can be inferred from Fig. 3b. Comparing for example the surface values with those at 300 m depth, the difference is clearly larger in these two regions than in the Tasmania region,

indicating stronger vertical gradients. We added in the revised version: "... and the stronger vertical gradient in those two regions, as indicated by the temperature and DIC profiles (Fig. 3b)"

L202 "a pattern influenced by sporadic but intense carbon uptake events (Fig. 3).": can you elaborate on how one can conclude this from Fig. 3?

It is based on the distribution shown in the violin plots in Fig. 3c for the South of Tasmania region. The mean flux is positive (outgassing) or close to zero in the background regime, but the distributions show a wide tail toward negative values (strong carbon uptake). When the fluxes are integrated over time, these less frequent but intense uptake events dominate, resulting in a net negative (into the ocean) flux.

 L220 "it is worth mentioning that both anticyclonic and cyclonic eddies do exhibit episodes of CO2 outgassing (Fig. 3c).": can we see this in Fig. 3c? If so, how, can you elaborate?

Yes, this can be seen in the distribution of CO2 fluxes shown in the violin plots in Fig. 3c for the Brazil-Malvinas Confluence region. While the mean and most eddy fluxes are negative (uptake), the distributions also include some positive values, indicating episodes of outgassing. Similar outgassing events are also present in the background and periphery regimes.

 L292ff importance of eddy type: mention that there is compensation of negative and positive flux anomalies?

Yes (that is a very good point), it became clearer with your suggestion to report the anomalous contributions. We have now added the following clarification to the paragraph you mentioned:

- "... Anticyclones typically enhance carbon uptake or suppress outgassing, whereas cyclones more often reduce uptake or promote outgassing. This opposing behavior leads to a partial compensation between positive and negative flux anomalies, particularly at the basin scale, where enhanced CO2 uptake by anticyclonic eddies tends to be offset by reduced uptake within cyclonic structures. Such compensation dampens the net short-term contribution of mesoscale activity to the overall Southern Ocean carbon sink, despite strong local contrasts."
  - L295 "despite the smaller region they occupy": could you provide the number here (area coverage in %, you should have the numbers in your analysis), see comment above, to get a sense of to which extent the eddies enhance the flux.

Although the area covered by the mesoscale regimes is shown in Table 1, we did not include the values for the total Southern Ocean. Thank you for noting this omission. In the revised version, we have added the total area coverage (%) for the entire Southern Ocean, and this information is now explicitly mentioned in the Summary.

• Given that the simulation is a historical simulation (i.e., natural fluxes, with the anthropogenic perturbation of increasing atmospheric CO2 on top): curiosity: do you have a hypothesis if the flux change (reduced outgassing) is more due to enhanced anthropogenic carbon uptake, or suppression of outgassing of carbon rich subsurface waters? If not, could you discuss these aspects in the discussion? It is indeed a very insightful question. However, our analysis focused primarily on characterizing the mesoscale influence on  $CO_2$  fluxes rather than on assessing long-term changes due to the anthropogenic signal. Therefore, we are not able to directly address this aspect within the scope of the present study.

**Technical comments:**

- L27. the role of cyclonic and anticyclonic eddies Done
- L93: mention specifically that it is eddy-rich Done
- L236f: repetitive with lines 208-210 Corrected
- larger Figure labels, some not readable in when printed We increased the font size
- add sign convention to Figure description: negative values outgassing, positive ingassing Done
- Fig.2 b) take care of colorblind-friendlier choice of colormap and add whether AE or CE are blue or red Done
- AE are presented in blue colour, CE in red colours. This is a bit confusing, the other way around would be way more intuitively Done
- Fig. 1a: standard deviation vorticity Corrected
- Fig. 4f: Frequency Corrected
- L323: I suggest to add units to the variables. Units were added.
- L338f: pCO2ocean Corrected

Thank you, all technical comments have been addressed in the revised version of the manuscript.

**References**

Schütte, F., Brandt, P., and Karstensen, J.: Occurrence and characteristics of mesoscale eddies in the tropical northeastern Atlantic Ocean, Ocean Sci., 12, 663–685, https://doi.org/10.5194/os-12-663-2016, 2016.

**References**

Smith, T. G., Nicholson, S. A., Engelbrecht, F. A., Chang, N., Mongwe, N. P., and Monteiro, P. M.: The Heat and Carbon Characteristics of Modeled Mesoscale Eddies in the South-East Atlantic Ocean, Journal of Geophysical Research: Oceans, 128, https://doi.org/10.1029/2023JC020337, 2023.